# Gene Therapy in Hereditary Retinal Dystrophies: The Usefulness of Diagnostic Tools in Candidate Patient Selections

**DOI:** 10.3390/ijms241813756

**Published:** 2023-09-06

**Authors:** Mariaelena Malvasi, Lorenzo Casillo, Filippo Avogaro, Alessandro Abbouda, Enzo Maria Vingolo

**Affiliations:** 1Department of Sense Organs, Faculty of Medicine and Dentistry, Sapienza University of Rome, 00185 Rome, Italy; lorenzo.casillo@uniroma1.it (L.C.); enzomaria.vingolo@uniroma1.it (E.M.V.); 2Department of Ophthalmology, Fiorini Hospital Terracina AUSL, 04019 Terracina, Italy

**Keywords:** Leber congenital amaurosis, retinitis pigmentosa, Bardet Biedl syndrome, congenital stationary night blindness, usher syndrome, OCT, ERG, gene therapy

## Abstract

Purpose: Gene therapy actually seems to have promising results in the treatment of Leber Congenital Amaurosis and some different inherited retinal diseases (IRDs); the primary goal of this strategy is to change gene defects with a wild-type gene without defects in a DNA sequence to achieve partial recovery of the photoreceptor function and, consequently, partially restore lost retinal functions. This approach led to the introduction of a new drug (voretigene neparvovec-rzyl) for replacement of the RPE65 gene in patients affected by Leber Congenital Amaurosis (LCA); however, the treatment results are inconstant and with variable long-lasting effects due to a lack of correctly evaluating the anatomical and functional conditions of residual photoreceptors. These variabilities may also be related to host immunoreactive reactions towards the Adenovirus-associated vector. A broad spectrum of retinal dystrophies frequently generates doubt as to whether the disease or the patient is a good candidate for a successful gene treatment, because, very often, different diseases share similar genetic characteristics, causing an inconstant genotype/phenotype correlation between clinical characteristics also within the same family. For example, mutations on the RPE65 gene cause Leber Congenital Amaurosis (LCA) but also some forms of Retinitis Pigmentosa (RP), Bardet Biedl Syndrome (BBS), Congenital Stationary Night Blindness (CSNB) and Usher syndrome (USH), with a very wide spectrum of clinical manifestations. These confusing elements are due to the different pathways in which the product protein (retinoid isomer-hydrolase) is involved and, consequently, the overlapping metabolism in retinal function. Considering this point and the cost of the drug (over USD one hundred thousand), it would be mandatory to follow guidelines or algorithms to assess the best-fitting disease and candidate patients to maximize the output. Unfortunately, at the moment, there are no suggestions regarding who to treat with gene therapy. Moreover, gene therapy might be helpful in other forms of inherited retinal dystrophies, with more frequent incidence of the disease and better functional conditions (actually, gene therapy is proposed only for patients with poor vision, considering possible side effects due to the treatment procedures), in which this approach leads to better function and, hopefully, visual restoration. But, in this view, who might be a disease candidate or patient to undergo gene therapy, in relationship to the onset of clinical trials for several different forms of IRD? Further, what is the gold standard for tests able to correctly select the patient? Our work aims to evaluate clinical considerations on instrumental morphofunctional tests to assess candidate subjects for treatment and correlate them with clinical and genetic defect analysis that, often, is not correspondent. We try to define which parameters are an essential and indispensable part of the clinical rationale to select patients with IRDs for gene therapy. This review will describe a series of models used to characterize retinal morphology and function from tests, such as optical coherence tomography (OCT) and electrophysiological evaluation (ERG), and its evaluation as a primary outcome in clinical trials. A secondary aim is to propose an ancillary clinical classification of IRDs and their accessibility based on gene therapy’s current state of the art. Material and Methods: OCT, ERG, and visual field examinations were performed in different forms of IRDs, classified based on clinical and retinal conditions; compared to the gene defect classification, we utilized a diagnostic algorithm for the clinical classification based on morphofunctional information of the retina of patients, which could significantly improve diagnostic accuracy and, consequently, help the ophthalmologist to make a correct diagnosis to achieve optimal clinical results. These considerations are very helpful in selecting IRD patients who might respond to gene therapy with possible therapeutic success and filter out those in which treatment has a lower chance or no chance of positive results due to bad retinal conditions, avoiding time-consuming patient management with unsatisfactory results.

## 1. Introduction

Hereditary retinal dystrophies (IRDs) are a group of retinal diseases, in which a gene defect leads to progressive dysfunction of the photoreceptors, retinal pigment epithelium and inner retinal layers [1,2]. Diseases primarily affecting peripheral or mid-peripheral vision were historically labeled Retinitis Pigmentosa (RP) by Franciscus Donders in 1855 [3].

To date, over 270 genes have been known to manifest different types of inherited retinal dystrophies, most of which have long belonged to the RP spectrum. Thanks to knowledge in the field of genetics, of which there are already over 60 genes described as causing RP, which manifest themselves with different phenotypes, several forms of RP belonging to the IRD spectrum are now known [3]. New genetic technologies, such as next-generation sequencing (NGS) and single nucleotide polymorphism (SNP) microarrays or comparative genomic hybridization (CGH), have increased the possibilities of identifying genes involved in IRDs [4,5,6,7].

Gene integration therapy offers great promise in treating these blinding conditions. In gene replacement therapy, viral or non-viral vectors introduce a wild-type copy of the pathogenic gene into the target retinal cells of interest. These therapies are designed to slow disease progression and attempt to restore visual function. Indeed, outer retinal photoreceptors and retinal pigment epithelial (RPE) cells are the primary targets in gene therapy.

Furthermore, the classification of RP as a disease primarily affecting peripheral or mid-peripheral vision has been improved by a more detailed description of the disease using sophisticated imaging and clinical, functional tests [3] and knowledge of the specific associated genetic mutation. Today, different forms of retinal dystrophies, such as LCA, BBS, CSNB and USH, are mainly caused by mutations in other genes. Still, sometimes, the same mutation can occur in different dystrophies, usually depending on the site of the transformation. This results in overlapping clinical symptoms in IRD that have been genetically classified as distinct.

For example, the RPE65 gene encodes a 65 kD retinoid isomerase expressed in the RPE. This protein is vital to the retinoid visual cycle because it converts the all-trans retinyl ester to 11-cis-retinol. When RPE65 is defective or absent, 11-cis-retinol is depleted, causing photoreceptor dysfunction. The mutation involving the RPE65 gene is c.499G>T, p.(Asp167Tyr). This variant has been reported with a frequency of 0.000026 in the European (non-Finnish) population [8]. The variant affects a highly conserved amino acid in the carotenoid oxygenase domain of the protein.

The same RPE65 gene is also involved in the development of the following retinal dystrophies: Leber Congenital Amaurosis (LCA), Retinitis Pigmentosa (RP), Bardet Biedel (BBS), Congenital Stationary Night Blindness (CSNB) and Usher Syndrome (USH). These might potentially be treated in the same way; therefore, for scientists, they represent an attractive option for gene therapy due to the relative delay in the onset of retinal degeneration in less severe diseases, despite the early onset of visual loss, thus providing a large treatment window in which cells are available for ‘rescue’.

Unfortunately, there are several concerns in applying gene treatment for these forms of IRD, mainly for better baseline clinical conditions than LCA, and this means that possible “side effects or negative reactions” may have worse results than expected. To exit from the impasse to treat or not treat, several inquiries and considerations were taken, but none of them aided the researcher in finding a straight way of treating patients using the golden rule.

Considering our experience in this review, we focus on the instrumental morpho-functional approach for classifying the suitability of gene treatment in different forms of IRDs, starting from those with RPE65 gene mutation, who are presently suitable for gene therapy.

In this family of IRDs with the same mutation in the same gene, there can also be significant variability in the expressivity of certain conditions, which is why clinical examinations, such as OCT, ERG, and visual field, were performed to obtain a clinical classification that complements the search for the residual retinal activity and potential photoreceptor rescue after treatment, to formulate a correct diagnosis to make clinical therapeutic decisions that can benefit these patients.

## 2. Epidemiology

Leber Congenital Amaurosis (LCA), which falls within early-onset retinal dystrophy (EORD), typically manifests in the early years of an individual’s life, often before reaching the age of one. The prevalence of LCA/EOSRD ranges from approximately 1 in 33,000 to 1 in 81,000 in the Caucasian population, and it is estimated to account for at least 5% of all inherited retinal diseases (IRDs). The known genes associated with LCA/EOSRD explain around 70% to 80% of the cases. The transmission is autosomal recessive. Only in rare cases do mutations in the CRX or IMPDH1 genes have an autosomal dominant inheritance pattern, which overlaps with the diagnosis of LCA [9,10].

Non-syndromic retinitis pigmentosa, a condition characterized by the degeneration of the retina, is estimated to have a global prevalence of approximately 1 in 5000 individuals (1 in every 4000 people in the United States), with carriers numbering around 1 in 1000 worldwide. Males are slightly more prone to be affected due to the X-linked form of the condition being expressed more frequently in males. In contrast, syndromic retinitis pigmentosa is much rarer, with Usher syndrome being one of its notable variants [11,12].

Congenital Stationary Night Blindness (CSNB) encompasses a set of retinal disorders that are non-progressive in nature. These conditions are characterized by difficulties in night or low-light vision, night or dim-light vision, disturbance or delayed dark adaptation, poor visual acuity, myopia, nystagmus, strabismus, normal color vision and fundus abnormalities. The exact prevalence of this condition remains uncertain; it has an estimated prevalence of 3–6 per 100,000 in the global population and is clinically and genetically heterogeneous. Inheritance patterns for CSNB can be autosomal dominant, autosomal recessive or X-linked recessive, as outlined in reference [13].

Bardet–Biedl syndrome (BBS) has a prevalence of approximately 1 in 125,000–160,000 in Europe. The cardinal features of BBS include retinal dystrophy, obesity, dystrophic extremities, renal structural abnormalities and male hypogenitalism. It is important to note that there is significant variation in the phenotype of BBS, both within and between families [14].

Usher syndrome is a rare genetic condition that affects people worldwide, with an estimated global prevalence (North European, Ashkenazi and Middle Eastern) of approximately 1 in 6000 to 1 in 30,000 individuals. The occurrence of Usher syndrome can vary across different populations and ethnic groups. The Caucasian population, specifically, has a prevalence of around 1 in 20,000 individuals. This syndrome contributes to 3–6% of childhood deafness cases and approximately 50% of combined deaf–blindness cases in adults. Among the different types of Usher syndrome, types 1 and 2 are the most commonly encountered, accounting for approximately 90–95% of all cases [15].

## 3. Gene Sequencing

Identifying the genetic origins of rare Mendelian disorders is growing in significance, following promising outcomes with gene-based therapy. Currently, widely performed genetic tests for LCA and other IRDs involve a microarray analysis that examines several known variations in predefined specific genes [16]. While this method is a suitable initial screening, it falls short in detecting novel mutations, proves costly in routine settings and exhibits a detection rate that varies depending on the population [17,18,19].

Secondary genetic tests include denaturing high-performance liquid chromatography and Sanger sequencing, predominantly targeting a subset of genes due to cost constraints and workload considerations [20].

Thanks to their remarkable speed and the continuous reduction in the costs of sequencing, massively parallel sequencing (MPS) technologies have emerged as an ideal choice for the molecular diagnosis of genetically heterogeneous disorders [21].

Two primary sequencing strategies for large-scale analysis are whole genome sequencing (WGS) and whole exome sequencing (WES). WES is widely employed for LCA; modern quantitative PCR (qPCR)-based enrichment strategy encompasses all exons of 16 identified LCA genes [22]. WGS has not been as extensively used as WES for mutation detection, largely due to cost-related factors, as WGS is at least twice as expensive as WES [23,24]. WGS has the capability to identify pathogenic structural variations originating in introns. Thus, the advantages of WGS extend beyond identifying non-coding pathogenic variations and, considering its more comprehensive exomic coverage, make it superior to WES. Consequently, PCR-free WGS should be considered the most comprehensive second-tier genomic test [25,26].

## 4. Treatment Available

Gene augmentation therapy is the main source of potential treatment for LCA. Studies involving genes commonly implicated in LCA, such as GUCY2D, RPE65, AIPL1, RPGRIP1, LCA5, CEP290 and RDH12, mainly using adeno-associated virus (AAV) or lentiviral vector-mediated gene-enhancement therapy, have shown more profound advances in the treatment of RPE65 mutations [27,28,29,30] by Luxturna™ (Voretigene neparvovecryzl; Spark Therapeutics, Philadelphia, PA, USA), which is the first drug treatment to have undergone a phase III study and is approved by the US Food and Drug Administration in gene therapy for LCA [29]. The active ingredient of Luxturna™ is a recombinant adeno-associated virus (AAV), with a therapeutic gene sequence that enables RPE cells to produce the retinoid isomerohydrolase RPE65. AAV particles have a size of approximately 25 nm and are stable in aqueous solution. Luxturna™ is an AAV designed to be administered to the subretinal space via the pars-planar approach. Introducing the corrected cDNA of the RPE65 gene together with other regulatory DNA sequences (including promoter, poly-A sequence) enables the restoration of RPE65 protein deficiency in patients with LCA with biallelic RPE65 mutation. The RPE treated with this drug produces the functional enzyme RPE65, generating 11- cis-retinal as part of the visual cycle. It is transported to the outer segments of the residual photoreceptors and is able to generate the light-sensitive photopigment and initiate the phototransduction cascade after exposure to light [31]. It is currently assumed that photoreceptors whose function has been restored do not degenerate further [32,33].

Improvements in visual acuity, perimetry and retinal sensitivity were reported and maintained up to 4 years after treatment in the latest clinical trials [34]; however, ongoing retinal degeneration was still observed with extended follow-up [27,28].

Among the criteria to be fulfilled in order to undergo gene therapy with Luxuturna, the patient must have received clinical confirmation of the diagnosis of retinal dystrophy, be a carrier of homozygous or biallelic heterozygous compound variants in the RPE65 gene and mutations should generally be classified as probable pathogenic variants according to the American College for Medical Genetics and Genomics (ACMG) classification. In addition, in order to ensure therapeutic benefit, the patient must possess a residual number of photoreceptor cells that are targeted for therapy; both RPE photoreceptor cells and nerve cells downstream of the retina must be present to produce an improvement, or desirable restoration, in visual function likely. Approval by the EMA leaves it to the treating physician to assess the existence of still-functioning retinal tissue; three criteria have been identified for its determination: total retinal thickness >100 µm at the posterior pole, a residual island in the central visual field (30°) (Isopter III4e) and an area without atrophy of at least three disc diameters [25,35].

There are also criteria that the practitioner must fulfill in order to be able to proceed with this therapy: experience in vitreoretinal surgery in children and young adults, experience in subretinal surgery in patients with advanced retinal dystrophy or other retinal-degenerative diseases and avoidance of accumulation of Luxturna™ outside the normal retinal space, in the vitreous cavity, in order not to incur unfavorable therapeutic effects, such as decreased bioavailability in the target tissue and/or increased systemic biodistribution [36,37]. During the application of the vector, it is advisable to use an operating microscope with intraoperative OCT and a vitrectomy system with the possibility of control of the injection rate by the surgeon [38]. With this instrument, it is indeed possible to differentiate the suprachoroidal, subretinal or intraretinal fluid in order to avoid accidental application of the drug in the suprachoroidal space instead of the subretinal space. It will also be possible to document the localization and extension of the subretinal injection. Before injecting Luxturna™, it may, therefore, be useful to open the potential subretinal space with a buffered electrolyte solution (e.g., balanced salt solution (BSS)). Luxturna™ can then be applied through the initial retinotomy. It is advisable to subsequently exchange air and fluid in order to remove any virus particles that may be present in the vitreous cavity. Care must be taken not to perform suction near the retinotomy. The surgeon decides on a case-by-case basis whether sclerotomy sutures and/or air tamponade are required [35].

And, in order to be able to perform such therapy, a number of technical conditions concerning the characteristics of the drug itself, as well as adequate training in the surgical use of Luxturna™ by the manufacturer, must be met. The equipment for regular storage and preparation of the solution for injection is required (the active substance must be stored at temperatures below −60 °C until use, and the cold chain must be guaranteed and reliably documented, the drug can be prepared for use no earlier than 4 h and this procedure should be carried out under aseptic conditions and with a sterile working method in a class II microbiological safety cabinet) [35].

The entire surgical team must be trained: the application is performed according to the manufacturer’s specifications of the EMA-approved agent via subretinal injection of 0.3 mL of the vector suspension (dose 1.5 × 10^11^ “vector genomes” (vg)) as part of a pars-plana vitrectomy (e.g., 23 G or 25 G) with subsequent air tamponade. It is recommended to treat the patient’s eye at intervals of at least six days; in addition, proper information on the handling of the bioprotective level 1 substance must be ensured, as well as proper disposal of the viral solution and disinfection of the operating room surfaces in accordance with local regulations and the current recommendations of the Robert Koch Institute (the manufacturer has been obliged by the EMA to conduct training on the reconstitution and application of Luxturna™ as part of a risk management program (RMP); participation in such training is mandatory prior to use on patients). With regard to the care and follow-up of patients after Luxturna™ therapy, any side effects must be recorded in a registry study, and the treatment of complications is carried out by the treating physician who injected the drug. For patient monitoring, the clinical examination and visual function tests are performed under standardized conditions, and to correctly assess the success of the therapy, at least the best-corrected visual acuity, global retinal sensitivity (GST), as well as OCT and FAF images, should be taken prior to surgery and in the postoperative course [39].

The application of Luxturna™ requires a surgical procedure (vitrectomy) and a subretinal injection for administration. In phase 1 and phase 3 follow-up studies, the efficacy and safety of this treatment were tested [34,40], and the results showed a drug safety profile consistent with the vitrectomy and subretinal injection procedure. Further, 68% of the patients analyzed in the study by Jalil A. et al. had mild adverse events, and only isolated cases of serious adverse events, such as irreversible loss of vision, macular atrophy, endophthalmitis and increased intraocular pressure leading to optic atrophy, occurred; these adverse events appear to be attributable to the surgical procedure. There were no drug-related serious adverse events or severe immune responses [41].

The therapeutic results obtained in the patients under study are sometimes mixed. For some patients, excellent improvements in all visual parameters after treatment were observed: improvement in photopic visual acuity, a function mediated by the foveal cone; this could be a secondary consequence of the improved cone health achieved by the increase in the RPE65 gene. OCT reveals mild inflammatory macular edema two weeks postoperatively, with sparing of the photoreceptor layer, a likely immune response to the drug [42,43]. In this case, the macular edema resolves by temporarily supplementing oral steroids, and the photoreceptor layer of the fovea remains unaffected. The improvement in visual function is maintained at a 2-year follow-up, which clearly indicates gene therapy’s vast potential in IRDs [41].

However, the postoperative follow-up of the second eye of these patients showed worse results; following an improvement in light sensitivity, a significant decline in visual acuity was noted two weeks after surgery. These patients presented with marked macular edema with photoreceptor involvement and loss, despite the immediate increase in oral steroids showing a good reduction in macular edema. The presence of worse macular edema in the second eye indicates previous immune sensitization responding to steroids, pointing to an immune-mediated etiology [41].

Further studies are ongoing in the area of gene therapy implementation. In these cases, increased immunomodulation for the second eye by using a higher and longer course of oral steroids could be considered if an inflammatory response is observed in the first eye. Overall, the future of gene implementation is promising. However, further studies must be conducted to investigate the potential adverse effects of the body’s immune responses to a ‘foreign vector’ to apply therapy while minimizing the associated risks safely [41].

## 5. Morphofunctional Finding in IRD

This review selected articles describing the characteristics of instrumental OCT and ERG tests performed on patients with a definite diagnosis, such as the RPE65 gene mutation, in LCA, RP, CSNB, BBS and USH. The characteristics described for each category of IRDs are a clear example of how these tests can provide an objective instrumental morphological assessment, from which a clinical classification ancillary to the genetic classification of IRDs can be obtained to select suitable patients for targeted gene therapy.

Twelve clinical studies reported morphofunctional characteristics of the IRDs [8,14,44,45,46,47,48,49,50,51,52,53]. Patients were divided into five groups according to their IRD.

Group 1 LCA patients;Group 2 RP patients;Group 3 CSNB patients;Group 4 BBS patients;Group 5 USH patients.

Table 1 shows OCT patterns in different IRDs.

Table 2 shows all the morphofunctional assessments in LCA, RP and BBS.

Table 3 shows ERG patterns in different IRDs.

### 5.1. Group 1: Leber Congenital Amaurosis (LCA)

This clinical entity is characterized by a clinical history that begins to present at birth.

Table 1 shows an OCT profile with a completely preserved photoreceptor layer and preserved inner retina (optical coherence tomography) in the spectral domain.

In Table 2, ERG tests show immediate extinction in all its scotopic and photopic components, as described in the literature [43].

Table 4 and Table 5 summarize the data from three studies concerning the characteristics of OCT and ERG in the LCA, respectively.

The first clinical study by Samuel G. Jacobson et al. recruited 24 patients with LCA of 10 known genotypes, ranging in age from 3 to 25 years, and it was studied clinically and using optical coherence tomography (OCT). Comparisons were made between OCT results across the horizontal meridian (60° central) of the patients. Three models were identified. The first model consisted of six patients with LCA having two different genotypes: the first genotype showed, for three patients, an easily identifiable retinal structure, with the presence of an almost normal outer nuclear layer (ONL) across the central retina. In contrast, the three patients in this group with different genotypes showed a severely dysplastic retina. The second model included 14 patients with five different genotypes but a common pathogenetic mechanism. These patients show well-formed foveal architecture but only residual central islands of normal or reduced ONL thickness. The third model included six patients with three different genotypes; of these, five patients, three with a different genotype from the other two, showed loss of central ONL or dysmorphology, suggesting early macular disease or poor foveal development [44].

The second study, by Lagan Paul et al. [45], examined a pair of siblings diagnosed with LCA aged eight years. Spectral-domain optical coherence tomography (SD-OCT) depicted subfoveal thinning with RPE atrophy in the RE; ERG examination showed a significantly reduced photopic and scotopic response [45].

In the third study, Testa F et al. [46] reported a multicenter longitudinal retrospective study of a total cohort of 60 Italian LCA patients with IRD associated with RPE65, studied with OCT and ERG, of whom only 43 patients with a mean age of 27.8 ± 19.7 years were included. OCT performed on 31 patients showed no cystoid macular oedema, macular holes or vitreomacular traction, while seven eyes of five patients (16.1%) showed epiretinal membranes. The thinning of the outer nuclear layer was the most common alteration (19 eyes (79%)), while the outer limiting membrane appeared disrupted in 42% of cases. The ellipsoid zone was more frequently altered in the extrafoveal areas (16 eyes (67%)) than under the fovea (13 eyes (54%)), and a minority of eyes showed signs of RPE atrophy (8 eyes (33%)). The central foveal thickness was found to significantly decrease with age at an average annual rate of −0.6%/y (*p* = 0.044). The ERG was examined in 34 patients. Undetectable scotopic and photopic responses were observed in 29 patients (85.3%) and reduced scotopic and photopic responses were seen in 5 (14.7%) [46].

### 5.2. Group 2: Retinitis Pigmentosa (RP)

The OCT Table 1 profile in the RP forms of the present study shows an alteration in the photoreceptor layer in the periphery, with changes also in the inner retina, like edema or wrinkling of the internal limiting membrane (ILM) [54]. The study of Milam et al. (1998) described that the first histopathological change in the RP is consistent with the shortening of the outer segments of the photoreceptors [54].

The ERG Table 2 test shows an extinction in the scotopic component that is less altered in the photopic component, which is preserved, although with reduced amplitude.

The visual field shows pericentral narrowing in a telescope shape, as described in the literature [55].

Only four studies concerning RP were identified for inclusion in this review.

Table 6 and Table 7 summarize the data from four studies conducted on RP patients assessed using OCT and ERG.

In the first study by Sanne K. Verbakel et al. [47], more than half of the patients presented macular abnormalities on OCT, with cystoid macular oedema (CME) being the most common finding, followed by epiretinal membrane formation, vitreomacular traction syndrome and macular hole. The same study found abnormalities in the ERG, such as delayed, diminished or absent scotopic responses, with subnormal a-wave on the dark-adapted ERG light flash. The response to flicker stimuli at 30 Hz is delayed and reduced. Oscillatory potentials are reduced in some patients with RP. The annual rate of decay in the full-field ERG among patients with RP ranges from 9 to 11 percent [47].

The second study, by Jin Kyun Oh et al. [48], showed reduced inner retinal thickness on OCT examination, resulting from the thinning of the GCL and IPL layers. A decrease in retinal thickness is also observed in the outer layers where foveal OS estimates the loss of photoreceptors. Finally, a reduction in the ellipsoidal zone (EZ) band has also been described [50].

The third study, by Mirjana Bjeloš et al. [8], reported the case of a 40-year-old man suffering from RP with the RP65 genetic mutation who had allelic variants that are less described in the literature. The morphofunctional features examined using spectral domain OCT (HRA+ OCT Spectralis^®^) show complete atrophy of the outer retina, loss of photoreceptors and a disruption in both the retinal pigment epithelium and the underlying choriocapillaris. The ERG shows extinguished scotopic and photopic responses [8].

A review by Hiram J. Jimenez-Davila et al. [49] reported the morphofunctional changes most commonly seen in RP patients: on OCT, there is disorganization of the outer retinal layers that progresses to a reduction in the outer nuclear layer with complete loss of both the outer segment and outer nuclear layer in the later stages of the disease, accompanied by inner retinal layers that remain relatively well preserved. Macular changes most commonly seen in these patients are CME, macular cysts and macular holes. The ERG data in these show reduced amplitude of the rods, maximum responses, oscillatory, conical and flicker responses. Subsequently, prolonged B-wave implicit times are observed. The possible involvement and loss of the cone photoreceptors lead to the reduced amplitude of the photopic, maximal and 30 Hz flicker responses. In the advanced stages of the disease, ERG responses may be completely extinguished [49].

### 5.3. Group 3: Congenital Stationary Night Blindness (CSNB)

OCT showed that the retinal morphology is preserved with a reduction in photoreceptors in the periphery (Table 1).

ERG Table 2 is absent in the scotopic component but present in the photopic component, with a reduction in the maximum ERG value. These results are in agreement with previous animal studies, indicating that even though the ERG of the rods is absent in the scotopic form, the ERG of the cones is normal [56,57,58].

Visual field tests cannot be performed in these patients due to nystagmus. In this group of patients, some may benefit more than others from Luxturna gene therapy, based on the morphological and electrophysiological features mentioned above.

Only one study concerning CSNB was identified for inclusion in this review.

Table 8 and Table 9 summarize the OCT and ERG characteristics of the study by Angela H. Kim et al. [14] of patients with CSNB.

Seven male patients with a mean age of 17.9 years were examined, the age of the patients ranged from 7 to 28 years and these patients were affected by CSBM caused by six different genetic mutations. Imaging obtained via OCT showed four patients with normal retinal morphology, two patients had globally reduced retinal thickness and only one patient had dome-shaped retinal architecture in the macular area. ERGs were extinguished to electronegative with photopic responses in the normal range for two patients; four patients had reduced b-wave amplitudes, with reduced ERG response. The b-wave amplitudes in the LA 30 Hz flicker were in a normal range. One patient did not undergo this method due to his young age [14].

### 5.4. Group 4: Bardet–Biedl Syndrome (BBS)

OCT showed that the external structure of the subfovea, including the myoid zone, ellipsoid zone and external limiting membranes, was completely absent (Table 1).

ERG (Table 2) was extinct in all its scotopic and photopic components in infancy. These results are in agreement with previous studies, showing reduced responses of the cones and rods in the ERG of most patients. Extinguished responses were found in 89% of patients [15].

The visual field test is also not possible in these patients due to the presence of nystagmus. Luxturna therapy is not useful in this group due to the complete disruption of the photoreceptor layer.

Only three studies concerning BBS were identified for inclusion in this review.

Table 10 and Table 11 summarize OCT and ERG data from three studies conducted on BBS patients.

The first was a retrospective study by Grudzinska Pechhacker MK. et al. [50] on a patient cohort recruited from nine academic centers in six countries (Belgium, the United States, etc.), consisting of sixty-seven individuals with two genetic variants: BBS1 (*n* = 38; 20 female and 18 male patients); BBS10 (*n* = 29; 14 female and 15 male patients). OCT performed in these patients showed retinal thinning with associated atrophy in the central macular region, with relatively preserved photoreceptors outside this area. Data from ERG assessments were available for 51 patients (76%; *n* = 35 for BBS1 and *n* = 16 for BBS10). Non-recordable ERGs were found in seven patients with BBS1 (20% of patients with BBS1, mean age = 22 years) and in 5 out of 16 (31%) patients with BBS10 (mean age = 16 years). Of the recordable ERGs, among patients with BBS1, 22 cases (78%, 7.8–27 years) showed reduced responses for both cone and rod photoreceptors, while 6 cases (21%, 15.1–35.2 years) showed a phenotype with reduced responses of both photoreceptors but more so in the cones. There were 11 (69%) ERGs in BBS10; of these, eight patients (73%, 4–16.3 years) showed reduced responses of both cone and rod photoreceptors, while three patients had a phenotype with reduced responses of both photoreceptors but more in the rods. For these three patients with BBS10-COD, the light-adapted photopic ERGs (LA) were severely reduced at a mean age of 22.3 years (12–39 years), and the responses of the rods were normal [50].

In the second study, by Xiaohong Meng et al. [51], retinal morphofunctional characteristics were analyzed in 12 Chinese patients (4 women and 8 men) with a mean age of 20.75 years (range: 8–37 years) suffering from BBS. These patients had 17 genetic variants of BBS, of which 5 were already known, and 12 were new variants. All patients had typical retinitis pigmentosa phenotypes with unrecordable or severely impaired cone and rod responses to full-field flash electroretinography (ffERG). Most patients showed unremarkable responses in visual-evoked potentials (PVEPs) and multifocal electroretinography (mfERG), while their flash visual-evoked potentials (FVEPs) indicated residual visual function [51].

The third study, by Fadi Nasser et al. [52], was characterized by OCT and ERG, including a cohort of sixty-one German patients aged between 5 and 56 years with 51 different biallelic mutations. OCT images of all 20 patients with BBS10 showed atrophy of the photoreceptor layer and loss of photoreceptor cells, mostly together with wrinkles of the inner limiting membrane. Fifteen patients with the BBS10 mutation showed large macular atrophy in the foveal area associated with loss of the photoreceptor layer and diffuse atrophy of the RPE. In the fifteen patients with the BBS1 mutation, ONL thinning was most noticeable. Five patients with the BBS9 mutation showed loss of the photoreceptor layer and foveal atrophy. Three BBS9 patients showed photoreceptor cell layer loss and foveal atrophy. The ERG was switched off in 54 of 61 patients; 5 patients (2 BBS12 (BBS58, 16 years and BBS59, 21 years), 1 BBS10 (RCD768 29 years), 2 BBS3 (BBS44-I, 23 years and BBS44-I, 13 years)) showed a scotopic response. VEP was performed on 24 subjects; of these, 13 showed a good VEP flash response, and 10 patients showed a reduction in VEP amplitude [52].

### 5.5. Group 5: Usher Syndrome (USH)

The OCT performed in these patients shows centrally present photoreceptors, the presence of cystoid macular edema with preserved inner retina, hyperreflective inner limiting membrane and macular vitreous traction, as reported in Table 1. The ERG reported in Table 2 shows a more impaired photoreceptor function with the absence of scotopic and maximal ERG response, as found in studies in mutated mice, where light exposure (∼50% photobleach) completely suppressed the a-wave of the ERG [58]. The photopic ERG response is present but reduced.

Table 12 and Table 13 show the study’s results by Samuel G. Jacobson et al. [53], in which thirty-three USH1B patients (age 2–61) with the MYO7A mutation were studied using OCT and ERG.

The OCT scans show that in two patients, structurally normal retina and photoreceptors are present in large regions of the central retina. Normal retina and photoreceptors are limited to a small region around the fovea in one patient. Eight patients had a normal retinal rod structure, retaining the density of the rod–rod ratio: normal or near-normal cone. Other patients, in contrast, had normal photoreceptors limited only to the cone-dominated fovea and its immediate surroundings. In a subgroup of eight patients with a mean age of 6.9 years (range 3–11), a loss of 14.3% per year of ONL thickness was calculated. Electroretinograms were performed in 20 of the 33 patients; all were abnormal. Only one patient had detectable b-waves of the rod ERG, about 5% of the average normal amplitude. Cone flicker ERGs were detectable in 11 out of 20 patients, and these waveforms ranged from 1% to 10% of the mean normal amplitude [53].

## 6. Discussion

Due to advances in science, the study of IRDs consists of the yearly discovery of new mutations causing these diseases and, for this reason, the study for the implementation of targeted gene therapy is now being developed in clinical and preclinical models for many IRDs. 

In this review, we reviewed the literature to describe the morphofunctional differences in IRD patients with genetic mutations that are common to them, such as RP, CSNB, BBS and USH, which are related to the RPE65 mutation, in order to identify clinical and diagnostic biomarkers, useful for stratifying patients in need of early therapy to increase diagnostic and treatment chances for gene therapy, which, to date, is the only treatment that appears to slow photoreceptor damage.

The morphological results shown in Table 3, Table 4 and Table 5 were obtained by performing the OCT and ERG instrumental diagnostic examinations reported in the literature and defining the clinical features of IRDs related to the same gene mutation as is the case, for example, in IRD- RPE65.

In the diagnosis of IRDs, the correlation of retinal morphostructural features, which demonstrate impaired retinal function, in association with the search for the specific gene mutation, which identifies the specific IRD, is extremely important. The importance of this correlation is supported by the realization of an accurate stratification of patients who are candidates for gene therapy, in the context of specific IRDs that show variability in manifestation. This classification must not only be based on knowledge of the genetic defect but, above all, on the phenotypic characteristics expressed morphologically, obtained through an analytical description of OCT and ERG examinations in patients. Indeed, only by using correct patient selection will it be possible to make a correct diagnosis within such a broad and overlapping spectrum of clinical disease but, also and above all, to intervene effectively and early with gene therapy by having a positive prediction.

The Food and Drug Administration’s (FDA) landmark approval of voretigene neparvovec for RPE65-associated Leber Congenital Amaurosis (LCA) stimulated tremendous optimism regarding retinal gene therapy for other monogenic IRDs. Voretigene neparvovec (Luxturna^®^), a recombinant gene therapy based on adeno-associated virus vectors, delivers a functioning copy of the 65 kDa gene specific to the human retinal pigment epithelium (RPE65) in the retinal cells of patients with reduced or absent levels of RPE65 protein, offering the potential to restore the visual cycle. A single-dose subretinal injection of voretigene neparvovec administered in each eye is approved in several countries worldwide for the treatment of vision loss in adult and pediatric patients with hereditary retinal dystrophy (IRD), associated with a confirmed biallelic RPE65 mutation and sufficient viable retinal cells [35].

Thus, given the recent approval of the drug, one can consider RPE65-mediated IRDs as an example of retinal dystrophy with a definite genetic diagnosis but a highly variable phenotypic presentation in patients carrying this genetic mutation that can manifest itself in different types of IRDs. Currently, the genetic selection criterion appears to be the main criterion for these patients. Considering Luxturna treatment, this criterion is the demonstration of bi-allelic disease-causing variants in RPE65. However, as with any genetic mutation, some gene variants may be hypomorphic or may have little effect on the activity of the protein resulting from the accumulated mutation, in which case, treatment with gene replacement may not result in a significant benefit, and, indeed, the risks and benefits of subretinal gene therapy must be carefully weighed. Indeed, in the context of the RPE65 mutation, the continued progression of retinal degeneration has been reported in both dogs and affected patients, despite evidence of efficacy after treatment with gene therapy [27,33]. These trends are indeed found in several gene therapy studies. Studies in the dog model after gene therapy with RPE65 have shown that long-term arrest of disease progression can only occur in retinal regions with relatively preserved photoreceptors at the time of treatment [59]. It is possible that some patients included in clinical trials of RPE65 gene therapy may have experienced locally severe photoreceptor losses above the threshold necessary to halt degeneration, affecting long-term efficacy results [27].

Therefore, before embarking on gene therapy for any type of mutation, it will be extremely important to further examine the morphofunctional retinal state in patients in order to identify the conditions that maximize the effectiveness of therapy, by searching for the presence of photoreceptors with even minimal residual activity and in order to achieve therapeutic success, which is based on arresting the degenerative process of the photoreceptors. 

To date, the detection of photoreceptors via detailed retinal imaging in patients with IRD still appears to have less emphasis as a pre-selection criterion that the genetic study. In the same way, within experimental therapy studies, clinical trial protocols are not always applied to define target retinal regions showing relatively preserved photoreceptors for treatment [45,60]. In fact, the assessment of the presence of photoreceptors in a given patient, particularly in end-stage disease, is crucial to define the actual presence of target cells on which gene therapy will have an effect [61]. This assessment should be carried out using clinical OCT and ERG tests, which, together with genetic diagnosis, should be considered prerequisites for selecting patients for therapy. The integration of morphostructural features allows for a clear definition of the highly variable phenotypes of IRDs. For example, it must be considered that total retinal thickness assessed via OCT may overestimate the presence of photoreceptors, as retinal thickening occurring as a result of internal cellular remodeling may mask external retinal loss [27]. Therefore, performing clinical tests for functional assessment of photoreceptors such as ERGs could be supportive in the evaluation of target cells for therapy.

For the same reason, even the use of fixed reference values of total retinal thickness, such as the thickness of the central subfield >100 μm considered for patients eligible for Luxturna therapy, should not be considered unequivocal proof of the presence of photoreceptors [27] but should always be used to assess their functionality. In fact, experimental evidence shows that degeneration can proceed even after patients have been treated, suggesting that there has been a certain threshold of photoreceptor loss in these patients [28,29,30,31,32,33,34,35,36,37,38,39,40,41,42,43,44,45,46,47,48,49,50,51,52,53,54,55,56,57,58,59,60,61,62]. For this reason, the loss of approximately 30% of the normal complement of photoreceptors is a therapeutic prerequisite with Luxturna, a threshold beyond which treatment will not have the desired outcome, with the progression of retinal degeneration occurring [33,59]. In fact, it is possible that locally preserved photoreceptors may continue to degenerate after treatment when the similarly treated neighboring retina has already exceeded the threshold for continued degeneration [62,63]. Another contraindication to Luxturna treatment is the presence of detectable photoreceptors that lack a relatively conserved supporting RPE, as the basic abnormalities in RPE65 -IRD, which are the primary cellular target of gene transfer, reside in this layer [64]. For this reason, performing SD-OCT cross-sections to determine the presence of detectable photoreceptors and whether RPE signals apical to Bruch’s membrane are still detectable [27] could be considered among the prerequisites for selection [27], always accompanied by the study of the functionality of these cells via ERG.

These clinical examinations should be taken into account when assessing the suitability for treatment of patients with any mutation developing IRD, as a functional retina may be indicative of the salvage potential of photoreceptors, particularly if a structural–functional dissociation is documented. Another factor to consider is the age of patients who are candidates for treatment. Indeed, to date, few gene therapy studies have been proposed in children who develop severe central dysfunction with amblyopia nystagmus and visual acuity below that of the mean RPE65 as a result of certain IRDs, as in the case of CSNB. Patients of a given age or disease severity should be considered for early intervention in children. Indeed, Luxturna therapy has recently been approved in children >1 year old [27,65]. In contrast, for adult patients with severe disease, demonstrating reliable, measurable vision mediated by classical photoreception should not be considered a fundamental therapeutic prerequisite, given the wide variability in results obtained in clinical trials, which shows that more careful retinal assessment is necessary [66,67].

### 6.1. Leber Congenital Amaurosis (LCA)

Retinal dystrophy LCA associated with genetic mutation RPE65 was the first to be explored for gene therapy. In December 2017, following a successful phase III randomized, controlled, open-label study, Luxturna [68] became the first FDA-approved ocular gene therapy for RPE65-associated retinal dystrophy [66,68]. Voretigene neparvovec was subsequently approved by all members of the European Union in 2018.

However, other genes may be responsible for the development of LCA. The RPE65 mutation is only responsible for about 5–10% of cases of LCA, so treatment with Luxturna will not be effective in other forms of LCA sustained by different genes [47].

The RPE65 gene is highly expressed in the retinal pigment epithelium, where it encodes the enzyme retinoid isomerase, which is essential for the production of the chromophore that forms the visual pigment in rod and cone photoreceptors of the retina. In LCA with congenital loss of chromophore production due to RPE65 deficiency, this is accompanied by progressive degeneration of the photoreceptors. Ref. [69] considered early childhood onset, characterized by severe and progressive vision loss, nystagmus, absence of a normal pupillary response and an almost absent electroretinogram (ERG). Franceschetti’s oculo-digital sign, which includes poking, pressing and rubbing of the eyes, is pathognomonic. The prevalence is estimated at 1:33,000 live births in the global population [70]. This is another example of a historical, phenotype-based diagnosis that has subsequently been shown to describe a group of genetically heterogeneous conditions.

A recent systemic review of the five prospective studies and one RCT RPE65 LCA on gene therapy showed that improvements in visual function outcomes only last up to two years after treatment [71]. The disease stage of the patient undergoing gene therapy is an important consideration for the application of future therapy. Indeed, efficacy results are variable because, despite treatment, some patients with advanced ACL continue to show retinal degeneration. This finding argues in favor of starting gene therapy early in retinal dystrophy before photoreceptor degeneration becomes irreversible [33]. There may be a threshold point beyond which target cells that undergo degenerative processes are no longer responsive to therapy. Indeed, in an initial phase I study, LCA children showed the greatest gains in functional assessments [64]. However, the assessment of retinal morphofunctional features with instrumental tests listed in Table 3 could complement the genetic diagnosis and select patients who are candidates to benefit most from gene replacement therapy. Indeed, the results of the phase III study showed similar improvements in young patients compared to older patients [66]. Ultimately, since retinal degeneration continues to progress slowly after gene therapy, it may be necessary to more accurately select patients belonging to a certain spectrum of IRD, in view of their genotypic–phenotypic incongruence, thus, not only via genetic analysis but also through the careful study of retinal features [33,72].

Regarding OCT in LCA patients, the following evidence in the literature shows the photoreceptor layer completely preserved and preserved the inner retina. The study of five genotypes of LCA patients showed, on OCT, the presence of a preserved central ONL island but a decrease with eccentricity, while the foveal ONL peak may be normal or reduced, and the retinal thickness was at the lower limit of normal or subnormal [44].

OCT images were observed in a study conducted by Reiko et al. An animal model of LCA and RP of mutated RPE65 mice, compared with histological and electron microscopy results and electroretinography (ERG) features [73] showed that the layer of the inner and outer segments of the photoreceptors was represented by a diffuse hyper-reflective zone that resembled that found by [74] in humans. In fact, the layer of degenerated IS and OS photoreceptors appeared to include diffuse hyper-reflective zones, resulting from the disorganization and vacuolization of the outer segment discs in the early phase and the variable size of the outer segment of the rods in the first months of life. Subsequently, they observed a progressive thinning of the outer nuclear layer, while the thickness of the IS and OS layer of the photoreceptors remained unchanged. These two qualitative changes could not be differentiated on OCT images [73].

The same result was shown in an Italian cohort study that confirmed that RPE65 mutations are associated with a common phenotype, such as severe rod dysfunction, detectable cone function, structural–functional dissociation, relative foveal sparing despite abnormal visual acuities, a paucity of pigmentary changes in the early stages of the disease, whitish deposits and a fundus-like appearance in the later stages of life that allow the clinical diagnosis to be confirmed [27,33,75].

Literature data on ERG testing in LCA patients show immediate extinction in all its scotopic and photopic components [43].

ERG in LCA RPE65−/− animal models showed severe impairment of the significantly lengthened a- and b-wave latencies. These findings indicate severe functional disturbances of the photoreceptors, which cannot be diagnosed via OCT, which instead shows relative preservation of the photoreceptor layer thickness [73].

The OCT and ERG characteristics discussed are described in Table 6 and Table 7.

Analyzing the morphological features emerging from the above clinical tests, we can hypothesize that patients with LCA related to RPE65 gene mutation, who present morphological features at ERG of extinction but preserved retina on OCT, may benefit more from gene therapy with Luxturna than patients who present retinal morphological features now deteriorated at both ERG and OCT. These clinical findings suggest that it is important to start therapy as early as possible in these patients before there is irreversible photoreceptor damage.

### 6.2. Retinitis Pigmentosa (RP)

Within the spectrum of RP-related retinal dystrophy, several gene therapy studies have been published for the replacement of specific mutated genes, such as RPGR MERTK, RLBP1 and PDE6B [27]. These studies have shown variable benefits in selected patients, again suggesting how the selection of the most suitable patients is crucial for the optimal success of said therapy [76].

Recently, the initial 6-month results of the first phase I/II clinical trial evaluating the safety and efficacy of a subretinally administered AAV8 vector encoding the RPGR gene (AAV8- coRPGR) were published, and, structurally, an increase in outer nuclear layer (ONL) thickness on OCT was observed in treated eyes [77].

MERTK gene mutation-associated RP, similar to RPE65-associated LCA, involves RPE dysfunction with mutations in MERTK implicated in RPE phagocytosis of photoreceptor segments [78].

In general, mutations in MERTK are rare, affecting less than 1% of RP patients from consanguineous families originating in the Middle East, Saudi Arabia, Spain and Morocco. The prevalence of MERTK mutations in the French cohort account for 2%, while in the isolated population of the Faroe Islands, a large MERTK deletion is responsible for 30% of the RP cases [79].

A phase I dose escalation study evaluated SR administration of rAAV2-VMD2-h MERTK in six participants with MERTK-associated RP (NCT01482195) [79]. Only one patient in this study maintained visual gain at a 2-year follow-up.

Retinitis pigmentosa (RP) encompasses several clinical conditions caused by a large number of genetic alterations that, alone or in combination, cause damage to the molecular processes necessary for the creation, storage, utilization or recovery of rhodopsin. Typical retinitis pigmentosa, also known as cone-radicular dystrophy (RCD), results from primary loss of rod photoreceptors, followed by secondary loss of cone photoreceptors [80]. Cone-radicular dystrophy (CRD), conversely, is characterized by retinal pigment deposits visible on fundus examination and localized predominantly in the macular region [10,80,81].

The mechanism of rod cell death varies depending on the mutated gene, and the speed of rod degeneration is an important prognostic factor, as cones do not begin to degenerate until almost all rods have been eliminated [82].

Degeneration of the rods, the most numerous photoreceptors (120 million), whose highest concentration density is in the mid-peripheral retina, causes night blindness (nyctalopia) and loss of the perifocal visual field.

The visual deficit and subsequent blindness in this disease result from degeneration of the cones, which are fewer in number (6 million) and are concentrated at the level of the fovea [83], which is why central vision remains in relatively good condition until the advanced stage of the disease. This explains why RP patients are often diagnosed after the second or third decade of life [47].

This sequence of events underlies the prevalent symptoms of RP: night blindness, tunnel vision and progressive loss of central vision and complete or near-complete blindness [47].

Some forms of RP are related to the RPE65 gene mutation that causes a severe form of hereditary cone-radicular dystrophy (IRD) [84,85,86] that produces gradual vision loss until complete blindness [81,82].

Retinitis Pigmentosa affects more than 1.5 million patients worldwide. RP is the most common hereditary retinal dystrophy (IRD), with a worldwide prevalence of approximately 1:4000 [10], although ratios vary from 1:9000 in the general population for all ages [87] to 1:750 in the Indian population [88], depending on geographical location [75].

A distinction is also made between ‘non-syndromic’ RP with vision loss alone, which accounts for 70–80% of people worldwide affected, and the ‘syndromic’ form when it occurs with a systemic disease [11].

The breakdown of the inheritance pattern shows rough estimates in the general population of 65% non-syndromic (20% autosomal dominant, 13% autosomal recessive, 8% X-linked, 24% isolated or unknown) [89], 17% syndromic (12% Usher syndrome) [90] and 5% Bardet–Biedl syndrome [91], 10% systemic and 10% other or unknown [83].

Morphological features concerning RP were studied in [92,93], who identified the most common findings on OCT: cystoid macular edema (CME) was the most common, followed by epiretinal membrane formation (MER), vitreomacular traction syndrome (VMT) and macular hole. The early stages of the disease are characterized morphologically by the disorganization of the outer retinal layers and outer limiting membrane. As RP progresses, the thinning of the outer segments is accompanied by a decrease in the thickness of the outer nuclear layer. The later stages of RP are characterized by the complete loss of the outer segment and outer nuclear layer [94,95].

In patients with RP, careful analysis of the CME architecture could aid in selecting patients responsive to gene therapy [10]. Indeed, cystoid spaces in RP mainly occur in the inner ONL nuclear layer but can also occur in the outer nuclear layer, outer plexiform layer and/or GCL (ganglion cell layer) [96,97].

A recent study by Strong et al. showed that patients with evidence of concomitant cystoid spaces in both the INL and ONL were more likely to respond to pharmacological intervention than those with only fluid in the INL. This would appear to be related to the proximity of the ONL to the RPE, the retinal layer in which changes occur due to genetic mutations accumulated in these patients, compared to the INL [97,98].

In some cases, in addition to a decrease in the thickness of the outer segments of the photoreceptors, a thickening of the inner retinal layers is observed, the cause of which is still unclear. The thickening observed in some cases is secondary to edema formation in the retinal nerve fiber layer or as a result of neuronal–glial retinal remodeling in response to thinning of the outer retina, a phenomenon also highlighted by Aleman et al. (2007) [99].

Instead, the later stages of RP are characterized by the complete loss of the outer segment and the outer nuclear layer [100,101,102]. In contrast, as shown by Aleman et al. (2007) [99], the inner retinal layers, including the inner nuclear and ganglion cell layers, remain relatively well preserved. For this reason, even in end-stage RP patients, performing OCT could be useful in recruiting patients who could benefit from possible gene therapies. A decrease in thickness of the outer photoreceptor segments may also be accompanied by thickening of the inner retinal layers; this could be related to edema formation in the retinal nerve fiber layer and/or neuronal–glial retinal remodeling in response to thinning of the outer retina [99]. For this reason, retinal thickness analysis should not refer to fixed cutoffs, as retinal architecture varies in patients depending on the genetic mutations present and the degree of mutation developed.

Furthermore, SD-OCT imaging may reveal outer retinal tubules [92] and hyperreflective foci in the inner nuclear layer, outer nuclear layer and/or subretinal space in patients with advanced disease and atrophy of the outer retinal layers. These hyperreflective foci may represent migrating RPE cells and appear to be related to the condition of the RPE layer [93]. The evaluation of these retinal changes concerning precisely the RPE via OCT could support the selection of patients for gene therapy. Betulla et al. analyzed the width of the retinal ellipsoidal zone (EZ) via OCT and showed that the reduction in the EZ line is consistent with the reported rates of change for full-field electroretinograms (ffERGs) in RP patients. Given the phenotypic heterogeneity of RP, quantifying the progressive loss of the EZ line width in different etiologies of RP on a gene-by-gene basis may be a key primary outcome measure for selecting patients who are candidates for gene therapy [100].

ERG results in RP patients showed a reduced scotopic response caused by a dysfunction of the rods with reduced amplitude and maximal, oscillatory, conical and flicker responses. ERG responses may be completely extinguished in the advanced stages of the disease [51].

In some cases, the possible involvement and loss of cone photoreceptors lead to reduced amplitude of photopic, maximal and flicker responses at 30 Hz.

ERG is considered the gold-standard modality for diagnosing RP, establishing the basic function of photoreceptors and monitoring the progression of RP. Indeed, ERG can detect photoreceptor dysfunction early, even with minimal changes in clinical examination or imaging modalities [51].

Visual field tests are important for establishing baseline function and monitoring disease progression. In the early stages of RP, visual field measurements show a variable loss of peripheral vision, which progresses to an annular scotoma consistent with the tunnel vision described in the later stages of the disease [80]. However, for the aforementioned review, the visual field test was not included among the clinical examinations determining the most accurate selection of patients for gene therapy precisely because of the typical telescopic characteristics of these patients, which does not allow for a complete assessment of retinal function.

The OCT and ERG characteristics discussed are described in Table 8 and Table 9.

RP is one of the most studied forms of IRD, as shown by several studies that exist in the literature.

Considering the clinical course and the morphological alterations of the retina, in which cells are progressively involved until they become completely dysfunctional, it is possible to hypothesize that some forms of RP, such as those linked to the RPE65 gene mutation, could benefit from gene therapy if carried out in patients at an early stage of the disease. Indeed, the therapeutic attitude for patients with early stages of dystrophy to date is more cautious as they are characterized by good visual acuity. However, further therapeutic efforts should be considered in these young patients, always aware of the risks of the current therapeutic application, which is carried out through subretinal injection of the product, following a standard three-way par-plana vitrectomy, creating a localized retinal detachment with the presence of fluid that is slowly reabsorbed [27] to slow down photoreceptor degeneration and preserve the functionality of residual photoreceptors before they undergo irreversible degeneration.

### 6.3. Congenital Stationary Night Blindness (CSNB)

In patients with CSNB, gene therapy has, so far, had no clinical application since, despite presenting signs and symptoms common to many IRDs, it is characterized by the development of primary visual deficits that do not allow for the development of vision, and, therefore, visual recovery is thought to be insufficient, even after replacement of the altered gene.

Congenital Stationary Night Blindness (CSNB) is an inherited retinal disease (IRD) that causes night blindness in childhood, with heterogeneous genetic, electrophysical and clinical features. This retinal dystrophy is characterized by the dysfunction of rod photoreceptors and impaired signal transduction between photoreceptor cells and bipolar cells [101,102]. Several altered mechanisms causing CSNB have been found to be dependent on the genetic mutation presented: defects in the visual signal pathway related to rod photoreceptors, rod bipolar cell synapses or retinoid recycling in the retinal pigment epithelium. In the area of ocular manifestations, CSNB patients manifest night blindness, myopia, strabismus and/or nystagmus. Nystagmus is described as pendular, discontinuous and oblique nystagmus with high frequency and low amplitude. Some of these symptoms are common to other forms of IRD, such as cone and rod dystrophies; therefore, an accurate diagnosis is essential to predict future visual outcomes and seek appropriate treatment.

According to a recent review, more than 300 mutations were identified in 17 genes involved in the development of CSNB [103]. The inheritance pattern among the general population was autosomal dominant (AD) in 2% of cases, autosomal recessive (AR) in 40% of cases and X-linked recessive in 58% of cases [104].

Recent studies conducted on the Saudi Arabian population have shown that mutation of the RPE65 gene is also present in this type of IRD [103]. In particular, in dogs with CSNB, a deletion of four homozygous nucleotides (AAGA) has been found in the wild-type RPE65 gene, generating a frameshift mutation, which causes a genetic mistranslation with a premature stop codon. The mutation causes retinal dysfunction and an accumulation of lipid vacuoles in the RPE. Studies in humans have identified that the mutation in this gene is responsible for the autosomal dominant form of CSNB [14].

Performing clinical instrumental tests in CSNB allows for more accurate characterization of the morphological features of the aforementioned retinal dystrophy, which also presents wide phenotypic variability. Retinal imaging with spectral-domain optical coherence tomography (SD-OCT) shows that retinal morphology is preserved with a reduction in photoreceptors in the periphery (Table 1). In patients with CSNB, there is dystrophy of the rod photoreceptors, which shows a variable pattern on clinical examination. The cohort study in these patients shows the following groups of findings: Patient 1 had an intact ellipsoid line (EZ) and normal retinal architecture in both eyes. Patient 2 had an intact EZ line in the right eye, and the SD-OCT in the left eye showed no obvious defects in the anatomical structure except for a thinner retina. Patient 3 had a normal SD-OCT. Patient 4 had an overall thin retina [104].

The ERG test in CSNB patients (Table 2) is absent in the scotopic component but present in the photopic component with a reduced maximum ERG value. In the literature, CSNB is classified into four types based on electroretinography (ERG) [105]. The Riggs type shows a reduced scotopic ERG in the dark-adapted response [106], while the Schubert–Bornschein type shows a characteristic electronegative ERG pattern [107]. The Schubert–Bornschein type is divided into two subtypes: a complete type and an incomplete type; the electrophysiological pathway of the visual cells explains this distinction. Photoreceptors transmit visual information to bipolar cells, which are second-order neurons. The signal transduction pathway is different in the various types of photoreceptors: rods only make contact with depolarizing bipolar cells (ON), creating ON visual pathways. Cone synapses have depolarizing DBCs and hyperpolarizing OFF bipolar cells [14].

Therefore, CSNB is classified as complete and incomplete and is caused by the dysfunction of ON or ON–OFF bipolar cells, respectively.

In the context of CSNB patients with the RPE65 gene, the clinical features of the autosomal recessive form of the disease must be considered, as the RPE65 mutation is located in these genes. This hereditary mode of CSNB manifests itself as a complete form. These patients have mutations in proteins distributed on postsynaptic bipolar ON cells, which are necessary for the depolarization of the cell. This is why, in this form of CSNB, there is an almost complete blockage of ON synaptic transmission from the photoreceptors to the bipolar cells in both the visual pathways of the cones and rods, while the OFF pathway is preserved intact [106].

As reported in the literature in patients with CSNB, the ERG is characterized by the specific variant of the Riggs type. Riggs-type patients (also known as type I) do not have a scotopic ERG, do not have an ERG of maximum stimulation in the bright field and do not have a rod–cone interruption on the dark adaptation curve. This electrophysiological pattern results in a molecular defect, predictably localized at the level of the rod photoreceptors [108].

The OCT and ERG characteristics discussed are described in Table 10 and Table 11.

The visual field test in these patients is unreliable due to the presence of nystagmus, which makes this test difficult to perform [109].

Although there are currently no human clinical studies concerning the use of gene therapy in patients with CSNB, previous work in mouse models has shown that such therapy administered with viral vectors can restore visual function [110,111]. However, although the clinical retinal features of these patients are, in some cases, morphologically good at OCT with partially preserved function at ERG, CSNB has not received much therapeutic effort compared to other forms of IRD due to its clinical course [112]. Due to the progressive course, most patients with CSNB develop poor vision during childhood, resulting in nystagmus before the age of two years, and subsequently develop amblyopia.

Patients with retinal dystrophy, such as RP, are not so limited in their therapeutic window because they initially have unaltered vision, with or without nyctalopia, which eventually progresses to visual field constriction and a loss of central vision.

Consequently, clinical trials whose aim is to prevent the progression of dystrophy and restore visual function may not apply to the aforementioned IRD as it leads to the development of amblyopia. However, CSNB also needs further investigation to explain the exact genotype–phenotype mechanism of some forms in which morphofunctional tests show partial preservation of retinal structures. With the above developments, further therapeutic efforts should be undertaken for these patients, and the clinical therapeutic target should be re-evaluated. In fact, the randomized clinical trial Pediatric Eye Disease Investigator Group (PEDIG) for amblyopia showed that visual function can even be improved in patients aged 13–17 years, with vision improvement greater than one line [113]. The Amblyopia Preferred Practice Pattern guideline published by the American Academy of Paediatric Ophthalmology in 2018 suggests the treatment of amblyopia up to 10 years of age [114]. The youngest patient injected for the Luxturna clinical trial was eight years old [115], with a milder subretinal surgical protocol, recalling that children aged >1 yr have recently been included in study protocols with Luxturna in other forms of IRD [116]. Given these data, gene therapy could also be envisaged for patients with CSNB not hitherto included in studies, as treating photoreceptors early in life, before amblyopia develops in these young patients, could represent a possibility of slowing down the development of dystrophy.

### 6.4. Bardet–Biedl Syndrome (BBS)

Bardet–Biedl syndrome (BBS) is a rare autosomal recessive hereditary ciliopathy with a prevalence of approximately 1:160,000 in European populations [117]. To date, 23 genes have been associated with BBS. BBS is pleiotropic, characterized by a broad spectrum of symptoms affecting multiple organ systems. The main features are retinal photoreceptor degeneration, obesity, polydactyly, renal abnormalities, genital abnormalities and intellectual disabilities. Photoreceptor degeneration with early macular involvement is a predominant feature, with symptoms appearing during the first or second decade of life [117,118]. It is characterized clinically by a compromised morphological pattern on OCT and an electrophysiological pattern with absent ERG. For this reason, no gene therapy studies have been undertaken to date in patients with BSS, as the retinal cells lack a target on which to act.

Mutations in BBS genes disrupt ciliary assembly and cause an incorrect interaction between the cilium and the outer segment of the photoreceptor, and studies show that in the Caucasian population, mutated BBS1, BBS5 and BBS10 genes cause the development of defective primary cilia [118,119,120,121]. Induced pluripotent stem cells (iPSCs) with BBS1 defects were able to differentiate into RPE65 cells expressing less pigmented RPE [122,123].

Furthermore, degeneration is also evident in the reduction in rod and cone responses in the ERG of most patients, as scotopic rod and cone responses are undetectable in most BBS patients [123].

In BBS syndrome, OCT shows the photoreceptor layer completely altered and the retina reduced in thickness (Table 1). Studies concerning OCT results in mouse models showed a thinning of the ONL, indicating a reduction in the number of photoreceptors and associated with retinal degeneration of 50% of the thickness in control mice until the ONL was no longer distinguishable with advancing age [55].

This early anatomical abnormality suggests how early treatment in humans may be optimal. These studies have shown that a knockout mouse model of BBS10 recapitulates the retinal degeneration of human patients and shares its characteristics. In the retina of Bbs10−/− mice, rods and cones are present from early life but are abnormal and degenerate over time. The 5 Hz flicker ERG, which cannot be recorded early and is adapted to light, offers a solid endpoint for therapeutic rescue studies. The fact that useful functional vision is possible in mice, even when ERGs are not recordable, suggests that rescuing even a small percentage of cones could prove useful in patients [55].

However, the performance of ERGs in human patients with BBS (Table 2) shows extinction in all their scotopic and photopic components, in agreement with evidence from the literature [14,124]. The lack of response in the ERG of the cones could result from a reduction in the amplitude of the electrical signal below the detection threshold on the ERG. The latter could be due to a reduction in the number of activating cones or a low amplitude of the electrical pulses of individual cones [125].

The OCT and ERG characteristics discussed are described in Table 12 and Table 13.

Specifically, among the forms of BBS, the most commonly represented form is BBS 10, which accounts for almost 25% of cases and is, therefore, a high-throughput target for treatment [126]. Furthermore, the phenotype of Bbs10−/− mice is similar to that of humans, making them an excellent model to use in preclinical studies of potential therapies [21].

The BBS10 retinal phenotype can manifest as cone-radicular dystrophy or even isolated cone dystrophy [54].

BBS10 in humans is a single nucleotide insertion that results in premature termination and loss of protein expression [127]; a completely non-functional BBS10 gene can lead to little or no BBSome formation, explaining the severe retinal phenotype of this type of IRD in mice and humans [124].

Indeed, gene therapy in RPE65 congenital BBS is not effective because patients carrying this mutation do not have sufficient target cells available to ensure therapeutic benefit with the restoration of a healthy RPE65 gene [126].

### 6.5. Usher Syndrome (USH)

Usher syndrome (USH) is a genetically heterogeneous group of autosomal recessive deafness–blindness syndromes characterized by the development of RP, sensorineural hearing loss and potential vestibular dysfunction. To date, hearing loss can be managed with cochlear implants. However, visual dysfunction currently has no treatment available. This syndrome accounts for 18% of all cases of RP in the general population considering different ages [127]. These mutations are hypothesized to interfere in the Usher protein network located in the photoreceptor connection region 3–5. Genetic heterogeneity is also a characteristic fact of USH. The focus has, therefore, shifted from gene discovery to the search for pathological mechanisms [128,129]. However, the prospect of clinical studies on treating these retinal diseases [130] necessarily also requires clinical clarification of the expression of the classes for the three clinical subtypes of USH [128]. Thus, instrumental diagnostic methods could help estimate the natural history of retinal dystrophy through cross-sectional data, thus being diriment in selecting the candidate patient to undergo gene therapy, selecting the most suitable timing of initiation of therapy to reap the greatest benefits. The characteristic symptom of this dystrophy is nyctalopia, caused by the loss of the outer segments of the rods. In the early stages of USH, the loss of the rods causes a ring-shaped scotoma in the middle periphery, which may progress to involve the periphery and macula. In the final stage, damage to the cone photoreceptors induces the loss of central vision [131]. USH is caused by mutations in many different genes, leading to the expression of three clinical subcategories, of which the clinically more severe Usher subtype 1 (USH1) presents at birth with extreme hearing loss, vestibular difficulties and early-onset slowly progressive RP. There are six disease-associated genes associated with USH1, of which mutations in MYO7A (myosin VIIA) are linked to the most common form of USH1, Usher 1B, in the Caucasian population (in the USA and UK population recently studied) [132].

The mutant phenotypes in the retinas of Myo7a mutant mice suggest deficiencies in the overall turnover of outer segment disc membranes as a mechanism underlying retinal degeneration in the Usher 1B subgroup [130,133,134,135]. Recent studies have described an additional mutant phenotype that may contribute to retinal degeneration in Usher 1B, related to loss-of-function mutations in the RPE65 gene [136]. RPE65 function is impaired in the absence of MYO7A, which is normally located in the retinal pigment epithelium (RPE) and is responsible for the reactions of the visual retinoid cycle. The results show that RPE65 is degraded more rapidly in Myo7a mutant mice, which is probably related to its incorrect localization, and this would also explain the finding of lower RPE65 levels in these animals. This provides a clear example of how Usher 1B may be the result of a combination of impaired, but not complete, loss of function in several critical RPE-photoreceptor cell processes in the presence of multiple mutations such as RPE65 that exacerbate the retinal dystrophic picture in these patients [58]. Also, other studies have demonstrated the same results: MYO7A is required for the normal localization and function of the visual retinoid cycle enzyme, RPE65. Unfortunately, at present, there are only laboratory studies conducted on Myo7a mutant mice, which would appear to be resistant to acute light-induced damage, as they possess lower levels of RPE65, and partially mislocalized to light. In fact, RPE isomerase plays a key role in the retinoid cycle, and in Myo7a mutant mice, it results in a more rapid degradation of RPE65, suggesting that these lower levels are caused by its mislocalization [137,138,139,140,141].

In a study by Samuel G Jacobson et al., preserved OCT scans detected peripheral temporal field islands until the late stages of RPE65-related USH1B disease [28]. Considering the well-preserved central retina with normal structure and function in many patients with USH1B [135], it could be hypothesized that focal treatment, by subretinal injection of the drug, could take place in the transition zones from normal to abnormal retina adjacent to the central retina [135]. Gene replacement therapy may be initiated in patients with USH1B early in the course of the disease, as patients are often diagnosed before structurally evident retinal degeneration. Like most other forms of USH, there is no mouse model with a retinal degeneration phenotype [13,133,137]. This is why we find non-invasive human studies in the literature on patients with well-defined genotypes to better understand the phenotypic expression of retinal degeneration [135].

Further results from the present study suggest that it may be even more prudent to initiate focal treatment studies in the peripheral retina, assuming subretinal delivery is used.

All patients with MYO7A in the study of Samuel G Jacobson et al. had severely abnormal ERGs. Detectable b-waves of the rod ERG were found only in one patient in the present study [53].

Analysis of retinal morphology via SD-OCT scans in the same study showed that age was not a good predictor of the extent of the disease, as the thickness of the photoreceptor layer in a large region of the central retina could differ significantly between patients of comparable ages [53]; furthermore, the results of the above study showed both severe photoreceptor losses in childhood as well as relative preservation in patients in the third decade of life. In the study by Jacobson SG et al. [53], comparisons were made between mutant alleles in mild versus severe phenotypes. From the data comparing the rod disease of patients in this cohort, the authors hypothesized that MYO7A null alleles might be associated with milder dysfunction and less structural loss of photoreceptors at ages when other genotypes show more severe phenotypes [53]. Thus, structurally normal retinas and photoreceptors could be present in large regions of the central retina or could be limited to a small region around the fovea, implying a centripetal component to progressive retinal degeneration.

In OCT scans, it was observed that some patients had a normal retinal structure extending well into the rod ring [138], and some patients had normal photoreceptors limited only to the fovea dominated by cones. Further information on the differences in the rods in patients with USH1B was obtained from cross-sectional images of the retina and measurements of the photoreceptor laminae vertically in the ONL layer. Commonly, in USH1B and other USH genotypes [134,135], there is a greater extent of photoreceptors in the upper macular area than in the lower ONL, which we attribute to the higher number of rods, leading to an apparently slower rate of degeneration in this region [53].

Longitudinal data in these patients indicated a progressive loss of rod function with a variable trend, as has been shown in other retinal degenerations [53].

The longitudinal OCT data in the present cohort could be described by a central retention of normal photoreceptors that actually corresponds to a rapidly progressive centripetal sweep of photoreceptors from the mid-periphery towards the central regions [135] during the early years of the life of patients with this dystrophy. The nuclei of the rod photoreceptors are the dominant contributors to the extrafoveal ONL thickness. Therefore, it can be expected that the vision of the rods (rather than the cones) is closely related to the retinal structure [53].

As the extent of the normal retinal photoreceptor laminae diminishes, the remaining extent is in the central regions that are normally relatively rich in cones (rod/cone ratio). From the second decade of life onwards, most patients retained only a central island of normal photoreceptor layer thickness that continued to shrink slowly. This phase of the disease is likely dominated mainly by the loss of the cone [53].

A literature review reported that the most common qualitative retinal abnormality found in OCT scans was damage to the outer layer in the macular area. Specific alterations included loss/rupture of the outer limiting membrane, rupture of the ellipsoidal zone and loss of outer segments. In the study of Vanda S Lopes et al., a general analysis of the photoreceptor and RPE status was performed, and the following results were observed: In the macular area, the loss of photoreceptors and RPE was 93.8% in both cases. At the same time, the analysis in the subfoveal area observed photoreceptor loss in 50% of cases and RPE damage in 21.9% [142].

Testa et al. (2018) retrospectively evaluated 42 patients with USH1 (mean age, 34.4 ± 17.0 years). OCT findings revealed the presence of macular abnormalities in 126 of 268 eyes (47.0%), with the most common abnormalities as follows: epiretinal membrane ERM (51 eyes; 19.0%), cystoid macular edema CMO (42 eyes; 15.7%) and vitreomacular traction VMT (38 eyes; 14.2%). Notably, CMO is more common in patients with USH1 than in patients with USH2, as reported in the literature [45,142].

Other studies in the literature have found wide variability in the prevalence of macular abnormalities. In the study by Grigoropoulos et al., the results of 21 affected patients showed CMO in 19% and ERM in 64.3% [143]. The study by Hagiwara et al. used OCT to study 323 patients with RP, detecting CMO in 34 eyes (5.5%) and ERM in only 4 eyes [13]. Between these two extremes, a cohort study by Triolo et al. [144], who used SD-OCT to examine 176 eyes of 90 patients with RP, found that the most common retinal changes were ILM thickening (118 eyes; 67%) and ERM (48 eyes; 27.3%). In that study, CMO and MPC were detected in only 12.5% and 18.2% of eyes, respectively [145].

In the study of Vanda S Lopes et al. (2011) [142], the most frequent alteration found on OCT was the presence of retinal abnormalities, such as retinal micro pseudocysts (MPC) and cystoid macular edema (CMO).

Kim et al. (2013) [145] evaluated a total of 266 eyes and found that vitreomacular interface abnormalities (VMIAs) were present in 42.7% of eyes with RP. In that study, VMIAs were significantly more common in eyes with CMO than in those without CMO (64.2% vs. 36.8%; *p* < 0.001); in contrast, the IS/OS junction was better preserved in patients without CMO. In the study by Triolo et al. (2013) [144], ILM thickening was observed in 67% of RP eyes and ERM in 27.3%. These authors found that IS/OS in the foveal region appears to be independent of the association with an obvious CMO, contrary to the report by Kim et al. (2013), who found that CMO was strongly correlated with IS/OS disruption [145].

A new finding in the study by Vanda S Lopes et al. (2011) [142] was the presence of ILM alterations—in the form of hyperreflective dots or ripples—that had not previously been associated with ERM or ILM thickening in patients with USH. The most common macular alteration found in the above study was outer layer damage, consisting of disruption/absence of IS/OS in >90% of eyes, absence of ELM in >80% and disruption of OS/RPE in more than two-thirds of patients. CMO was significantly correlated with alterations at the OS/RPE junction but not with ILM or IS/OS abnormalities [142].

The results of the aforementioned study by Vanda S Lopes et al. (2011) [142] suggest that USH1 may initially affect the outer retinal layers, particularly photoreceptors, and that alterations in these layers promote the development of retinal layer alterations, including the more common cystoid macular edema (CMO). It has, therefore, been hypothesized that photoreceptor degeneration may induce a non-specific inflammatory response, which leads to photoreceptor cell death.

The OCT and ERG characteristics discussed are described in Table 12 and Table 13.

The study of di Vanda S Lopes et al. (2011) [142] provided results on qualitative retinal changes detected via SS-OCT in a group of pediatric patients with USH1 associated with MYO7A. The results showed that the most common qualitative abnormality in the retina of the macular area was damage to the outer layers, including loss or disruption of the outer limiting membrane (ELM) (84.4% of eyes), disruption of the ISeZ (28 eyes; 87.5%) and loss of the OS (29 eyes; 90.6%). The damage to the different segments of the RPE was 93.8% PhaZ, 90.6% RPEmel and 0% RPEmitz. Following these results, the same study hypothesized that MYO7A retinopathy causes photoreceptor loss as the primary event and that the inner segments of the RPE are subsequently affected during disease progression. For this reason, even in retinal dystrophy that develops in USH1, monitoring the condition of the photoreceptors during follow-up using instrumental examinations such as OCT could be an important parameter to monitor in order to ensure adequate follow-up of patients with USH, allowing for early diagnosis and possible treatment of pathological changes in the retinal cells using gene therapy.

## 7. Conclusions

There are currently no studies in the literature that provide an accurate classification of Inherited Retinal Diseases (IRDs) that include an assessment of the mutated genetic profile alongside the varying morphofunctional patterns seen in these patients. These patterns result in different phenotypes depending on factors, like the extent of mutation, the number of mutations in different genes within an individual patient and how these mutations interact with the environmental DNA pattern.

Based on the data from this study, the group classifications can serve as valuable tools for distinguishing between Leber Congenital Amaurosis (LCA), Retinitis Pigmentosa (RP), Bardet–Biedl Syndrome (BBS) and Congenital Stationary Night Blindness (CSNB) within the context of IRDs. This distinction holds significance, especially considering the potential impact of gene therapy on altering disease progression and treatment strategies. Thus, further prospective studies are warranted to enhance our comprehension of the connections between genotype and phenotype in these disorders.

Genetic analysis alone requires supplementary instrumental support to clinically classify IRDs accurately and determine the most suitable candidates for gene therapy. Currently, the supporting instrumental examinations encompass electroretinography (ERG) and optical coherence tomography (OCT), which, respectively, examine retinal electrophysiology and morphology.

The integration of gene testing with clinical–instrumental examinations, specifically ERG and OCT, has yielded results that more precisely differentiate and categorize various forms of IRDs. This is especially pertinent due to the considerable variability in phenotypic manifestations, even between members of different IRD families who, however, carry mutations in the same RPE65 gene, enabling the correct classification and, thus, effective and safe gene therapy.

Consequently, it can be posited that greater emphasis should be placed on investigating the potential application of gene therapy in RP cases where there is a positive gene test, presence of ERG albeit reduced and the presence of photoreceptors in early stages of the disease before extensive retinal dystrophy sets in. Such an approach should be balanced with realistic expectations, cost considerations and the avoidance of false hope.

While more therapeutic studies have been conducted in cases of LCA, there is a call to explore gene therapy at the early stages of the disease, even in patients with positive genetic tests, absent ERGs but preserved retinal morphology on OCT. This holds the promise of longer-lasting results compared to current outcomes, particularly because LCA forms exhibiting an altered OCT retinal profile and altered ERG response may derive limited benefits from the aforementioned therapy.

Conversely, no therapy studies have been performed in CSNB, primarily due to the early onset of visual deficits and their association with amblyopia. Yet, for patients with positive genetic tests, diminished but still present ERGs, reduced peripheral photoreceptors on OCT and preserved retinal morphology, gene therapy could be contemplated to slow photoreceptor degeneration and allow for better visual development.

In the case of BBS, despite a positive genetic test, the clinical manifestation involves extinguished ERGs and diminished retinal thickness, with a severely disrupted photoreceptor layer on OCT. As such, gene therapy is not advisable given its lack of potential benefits.

Turning to USH (Usher syndrome) patients with an RPE65-related condition, it might be worthwhile to focus on targeted gene therapy studies for those who display OCT scans featuring intact photoreceptors and partially preserved residual photoreceptor function at early disease stages. These features are consistent across different ages, rendering these patients suitable candidates for gene therapy interventions.

## Figures and Tables

**Table 1 ijms-24-13756-t001:** Optical coherence tomography (OCT) classification in Leber Congenital Amaurosi (LCA), Retinitis Pigmentosa (RP) and Bardet–Biedl syndrome (BBS).

	Photoreceptor	External Retina	Internal Retina	ILM	Vitreo
Leber congenital amaurosis L.C.A.	Normal	Normal	Normal	Normal	Normal
Retinitis Pigmentosa	Presence in the center	EMC	Preserved	Hyperreflective	VMT
Congenital Stationary Night Blindness	Reduced in periphery	Normal	Normal	Normal	Normal
Bardet–Biedl syndrome BBS	Presence in the center	Preserved	Preserved	Hyperreflective	VMT
Usher Syndrome UHS	Presence in the center	EMC	Preserved	Hyperreflective	VMT

**Table 2 ijms-24-13756-t002:** Instrumental characteristics in Leber Congenital Amaurosi (LCA), Retinitis Pigmentosa (RP) and Bardet–Biedl syndrome (BBS).

	Erg Scotopic	Erg Maximal	Oscillatory Potential	Erg Fotopic	Flicker 30 Hz	Photoreceptors	Internal Retina	External Retina	ILM	C.V.
LCA	Extinct	Extinct	Extinct	Extinct	Extinct	Normal	Normal	Normal	Normal	not executable
RP	Extinct	Extinct	Extinct	Extinct	Extinct	Presence in the center	EMC	preserved	Hyperreflective	telescope pericentral narrowing
BBS	Hypovolted	Electronegative	Normal	Hypovolted	Normal	Present	Normal	Normal	Strongly altered	not executable

**Table 3 ijms-24-13756-t003:** Electroretinogram (ERG) classification in Leber Congenital Amauosi (LCA), Retinitis Pigmentosa (RP) and Bardet–Biedl syndrome (BBS).

	Erg Scotopic	Erg Max	Oscillatory Potential	Erg Fotopic	Flicker 30 Hz
Leber congenital amaurosis L.C.A.	Absent	Absent	Altered	Absent	Absent
Retinite Pigmentosa	Absent	Absent	Altered	Reduced	Present
Congenital Stationary Night Blindness	Absent	Reduced	Reduced	Present	Present
Bardet-Biedl syndrome BBS	Absent	Absent	Altered	Reduced	Reduced
Usher Syndrome UHS	Absent	Absent	Altered	Reduced	Reduced

**Table 4 ijms-24-13756-t004:** OCT features of LCA in each study (ONL outer nuclear layer; ELM external limiting membrane; NFL nerve fiber layer).

Authors	N° Patients	Age	Type of Study	Photoreceptor	External Retina	Internal Retina	ILM	Vitreo
Samuel G. Jacobson et al., 2015 [44]	24	13.83 ± 19.7	Clinical	Normal	Normal/subnormal	Not reported	Not reported	Not reported
Lagan Paul et al., 2020 [45]	2	8	case series and review of literature	Not reported	Not reported	hyperreflective dome shaped mass within the NFL	Not reported	Not reported
Testa F et al., 2022 [46]	60	27.8 ± 19.7	Retrospective longitudinal multicenter study	Not reported	ONL reduced ELM disrupted Ellipsoid zone altered in extrafoveal areas	Not reported	Not reported	Not reported

**Table 5 ijms-24-13756-t005:** ERG features of LCA in each study.

Authors	N Patients	Age	Type of Study	Erg Scotopic	Erg Max	Oscillatory Potentials	Erg Fotopic	Flicker 30 Hz
Lagan Paul et al, 2020 [45]	2	8	case series and review of literature	Reduced	Not reported	Not reported	Reduced	Not reported
Testa F et al., 2022 [46]	31	27.8 ± 19.7	Retrospective longitudinal multicenter study	Absent	Not reported	Not reported	Absent	Not reported

**Table 6 ijms-24-13756-t006:** OCT of RP in each study (ONL outer nuclear layer; MER epiretinal membrane; CME cystoid macular edema; EZ ellipsoid zone, OS outside segment; IPL inner plexiform layer, GLC ganglion cell layer).

Authors	N Patients	Age	Type of Study	Photoreceptor	External Retina	Internal Retina	ILM	Vitreo
Sanne K.Verbakel et al., 2018 [47]	Not reported	Not reported	Review	Not reported	MER	CME	Not reported	VMT
Jin Kyun Oh et al., 2020 [48]	206	Not reported	Review	44 P reduced	32 P OS reduced 162 P EZ reduced 10 P ONL reduced	12 P GCL reduced 12 P IPL reduced	Not reported	Not reported
Mirjana Bjeloš et al., 2022 [8]	1	40	Case report	Loss	Complete atrophy	Disruption	Not reported	Not reported
Hiram J. Jimenez-Davila et al., 2022 [49]	Not reported	Not reported	Review	Loss	OS reduced	CME	Not reported	Not reported

**Table 7 ijms-24-13756-t007:** ERG features of RP in each study.

Authors	N Patients	Age	Type of Study	Erg Scotopic	Erg Max	Oscillatory Potentials	Erg Fotopic	Flicker 30 Hz
Sanne K.Verbakel et al., 2018 [47]	Not reported	Not reported	Review	Reduced/absent	Reduced	Reduced	Reduced	Bright flash
Mirjana Bjeloš et al., 2022 [8]	1	40	Case report	extinguished	Not reported	Not reported	extinguished	Not reported
Hiram J. Jimenez-Davila et al., 2022 [49]	Not reported	Not reported	Review	Reduced	Reduced	Reduced	Reduced	Reduced/bright flash

**Table 8 ijms-24-13756-t008:** OCT features of CSNB in each study.

Authors	N Patients	Age	Type of Study	Photoreceptor	External Retina	Internal Retina	ILM	Vitreo
Angela H. Kim et al., 2022 [14]	7	17.9	cohort study	2P Reduced	4P Preserved 2P ellipsoid zone normal or reduced in some case	4P Preserved in some case 2P Reduced	Not reported	Not reported

**Table 9 ijms-24-13756-t009:** ERG features of CSNB in each study.

Authors	N Patients	Age	Type of Study	Erg Scotopic	Erg Max	Oscillatory Potentials	Erg Fotopic	Flicker 30 Hz
Angela H. Kim et al., 2022 [14]	7	17.9	cohort study	2P Normal 2P Absent, in some case reduced	Not reported	Not reported	2P Normal	2P Normal 2P Reduced

**Table 10 ijms-24-13756-t010:** OCT features of BBS in each study (ONL outer nuclear layer; ELM external limiting membrane; ILM internal limiting membrane).

Authors	N Patients	Age	Type of Study	Photoreceptor	External Retina	Internal Retina	ILM	Vitreo
Grudzinska Pechhacker MK.et al., 2021 [50]	Tot: 67	22.3	retrospective study	Presence out of the center	Disrupted	Reduced	Not reported	Not reported
Xiaohong Meng et al., 2021 [51]	Tot: 12	20.75	cohort study	Not reported	myoid zone absent, ellipsoid zone absent, ELM absent		Hyperreflective	Not reported
Fadi Nasser et al., 2022 [52]	Tot: 61	24.5 +/− 12.3	cohort study	23P Loss 5P Reduced	15P ONL reduced	23P ILM reduced 5P Reduced in foveal area	15P Hyperreflective	Not reported

**Table 11 ijms-24-13756-t011:** ERG features of BBS in each study.

Authors	N Patients	Age	Type of Study	Erg Scotopic	Erg Max	Oscillatory Potentials	Erg Fotopic	Flicker 30 Hz
Grudzinska Pechhacker MK. et al., 2021 [50]	Tot: 67	22.3	retrospective study	Absent Reduced in some patients: attenuated cone function		Not reported	Not reported	Not reported
Xiaohong Meng et al., 2021 [51]	Tot: 12	20.75	cohort study	Absent Minimal response in some patients [The ffERG was unrecordable in most of the patients, only patient (F3-II:1) had partial residual rod response.]	Not reported	FVEP mild delayed but visible P2 waves. Moderately reduced P2 wave amplitude	Not reported	Not reported
Fadi Nasser et al., 2022 [52]	Tot: 61	24.5 +/− 12.3	cohort study	Absent	Absent	Altered	Not reported	Reduced

**Table 12 ijms-24-13756-t012:** OCT features of UHS in each study.

Authors	N Patients	Age	Type of Study	Photoreceptor	External Retina	Internal Retina	ILM	Vitreo
Samuel G. Jacobson et al., 2011 [53]	33	14.85	cohort study	2P Normal 1P Normal nearby fovea 8P Normal rod (periphery) Normal rod (periphery)	2P Normal Normal nearby fovea	2P Normal Normal nearby fovea	2P Normal	2P Normal

**Table 13 ijms-24-13756-t013:** ERG features of UHS in each study.

Authors	N Patients	Age	Type of Study	Erg Scotopic	Erg Max	Oscillatory Potentials	Erg Fotopic	Flicker 30 Hz
Samuel G. Jacobson et al., 2011 [53]	20	14.85	cohort study	20P Absent	20P Reduced	20P Reduced	Not reported	11P reduced 9P Presence 11P Present

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
