# Peer review of "Gene Therapy in Hereditary Retinal Dystrophies: The Usefulness of Diagnostic Tools in Candidate Patient Selections"

_ijms, 2023, doi:10.3390/ijms241813756_

Round 1

Reviewer 1 Report

In the submitted article entitled “Gene therapy in hereditary retinal dystrophies, usefulness of 3 diagnostic tools in candidate patient selections”, the authors provide a thorough review of the current methods used to diagnose IRDs and evaluate the effectiveness of treatments, including discussion on the known genetic causes of important IRDs, and what is currently known about the effectiveness of gene therapies that have been used to treat IRDs. While I find this to be a thorough and well-researched review article, the writing has many grammatical errors and is difficult to follow. I had difficulty understanding some of the concepts being discussed because of the writing errors. Otherwise, I think that this has the potential to be an excellent article, once the writing has been thoroughly edited. I have not suggested specific edits because they are so numerous, but I suggest starting by cleaning up the Abstract, which has many grammatical errors and does not present a clear summary of the article.

I have included this assessment in my overall comments.

Reviewer 2 Report

The manuscript discusses the effects of using voretigene neparvovec gene therapy implementation by for RPE65‐associated Leber congenital amaurosis and other IRDs, which is an important research topic. However, the authors should improve the readability of the manuscript. There are many typos. I have several comments.

1.       MLE and MLI are somewhat weird abbreviations. Internal limiting membrane (ILM) might be a better one. MLI or ILM should be introduced in the manuscript where it is initially written.

2.   The discussion section seems to be well-written except few typos. However, the authors did not include some IRDs like Stargardt disease and X-linked retinoschisis other than the five IRDs discussed in the manuscript. Is the reason the authors want to classify IRDs clinically based on RPE65 gene mutation in the manuscript? Stargardt disease is caused not by RPE65 gene mutation but by changes in ABCA4 gene. Is it that the authors want to avoid discussing gene therapy candidates as Luxturna is the first and only FDA-approved gene therapy? On the other hand, the authors mention an Usher syndrome patient with MYO7A mutation anyway.

3.       ‘Associate AdenoVirus vector’ is not correctly used. The authors could use ‘Adenovirus-associated vector’ instead.

4.       Line 27, page 1. Instead of using ‘less rare incidence’, the authors are suggested to use possible synonyms like more frequent, more common or more prevalent. This is suggested for having a more fluent readability of the manuscript.

5.       The order of Table 2 and Table 3 is wrong in the captions/titles of the tables. One of the tables does not have any column titles at all. It is difficult to follow information present in tables possibly due to table format and typos. Could the authors make improvements to tables?

6.       What does EMC mean in Table 1?

7.       Typos in the manuscript.

a.       Typo in the abstract, ‘photoreceptor unction’ should be changed. Possibly into ‘photoreceptor function’.

b.       Page 13, Line 387, ‘RP65’ to ‘RPE65’.

c.       Typos of abbreviation. ‘CSBN’ should be changed to ‘CSNB’. Page 2, Line 77, Page 13, Line 414, and in more places in the manuscript. Also, Page 41, Line 421 has a ‘CSBM’.\

d.       What is ACL on Page 21, Line 627 and Line 632? Is it Leber congenital amaurosis?

Reviewer 3 Report

1.       Please kindly cite the following sentence based on a suitable reference, it has been known that over 270 genes have been known in manifestation of different types of inherited retinal dystrophies. “280 genes have been associated with various types of IRDs.”

2.       Page 2, line 77, what is BSS stand for, please clarify?

3.       All the prevalence rates should be addressed by adding more information in terms of the location and population characteristics; please do not report in general.

4.       Please kindly mention the type of your review.

5.       Please kindly provide more information regarding the other genetic subcategories of all IRDs that investigated in your study.

Minor English editing could be necessary.

Round 2

Reviewer 3 Report

All comments have been considered correctly.

Author Response

Dear reviewer, thank you for your positive comment.

I attach the revised manuscript for you, as requested by all reviewers of the prestigious journal. 

I remain at your disposal 
Best regards